# Characterization of the Dynamic Gastrointestinal Digests of the Preserved Eggs and Their Effect and Mechanism on HepG2 Cells

**DOI:** 10.3390/foods12040800

**Published:** 2023-02-13

**Authors:** Yan Wu, Xiujuan Li, Meihu Ma, Gan Hu, Xing Fu, Jihong Liu

**Affiliations:** 1College of Food Science and Technology, Huazhong Agricultural University, Wuhan 430070, China; 2National R&D Center for Egg Processing, Huazhong Agricultural University, Wuhan 430070, China; 3College of Science, Huazhong Agricultural University, Wuhan 430070, China

**Keywords:** preserved eggs, digestive characteristics, HepG2, apoptosis

## Abstract

Preserved eggs, an alkaline-fermented food, have been widely searched for their anti-inflammatory activity. Their digestive characteristics in the human gastrointestinal tract and anti-cancer mechanism have not been well explained. In this study, we investigated the digestive characteristics and anti-tumor mechanisms of preserved eggs using an in vitro dynamic human gastrointestinal-IV (DHGI-IV) model. During digestion, the sample pH dynamically changed from 7.01 to 8.39. The samples were largely emptied in the stomach with a lag time of 45 min after 2 h. Protein and fat were significantly hydrolyzed with 90% and 87% digestibility, respectively. Moreover, preserved eggs digests (PED) significantly increased the free radical scavenging activity of ABTS, DPPH, FRAP and hydroxyl groups by 15, 14, 10 and 8 times more than the control group, respectively. PED significantly inhibited the growth, cloning and migration of HepG2 cells at concentrations of 250–1000 μg/mL. Meanwhile, it induced apoptosis by up/down-regulating the expression of the pro-apoptotic factor Bak and the anti-apoptotic gene Bcl-2 in the mitochondrial pathway. PED (1000 μg/mL) treatment resulted in 55% higher ROS production than the control, which also led to apoptosis. Furthermore, PED down-regulated the expression of the pro-angiogenic genes HIF-1α and VEGF. These findings provided a reliable scientific reference for the study of the anti-tumor activity of preserved eggs.

## 1. Introduction

Hepatocellular carcinoma (HCC) is one of the six most common malignancies worldwide and the second leading cause of death from cancer [1]. Currently, the common treatment methods include radiotherapy, chemotherapy, surgical resection and liver transplantation. However, their treatment efficacy is not satisfactory, with a recurrence rate of 70%. Moreover, adverse reactions such as chemotherapy drug resistance, immunosuppression and immunological rejection can also occur after such treatments [2]. Therefore, functional foods with anti-tumor activity have gained a great potential for development.

Preserved eggs, a traditional Chinese food, are also known as “thousand-year eggs”, “century eggs” or “millennium eggs”, and are popular among consumers in countries such as Vietnam, Korea, Thailand, the Philippines and Singapore [3]. They are rich in protein, fat lecithin and nutritional elements such as iron, calcium, magnesium and zinc [4]. Moreover, the preserved eggs had a wide range of biological effects, such as anti-inflammatory, anti-tumor and blood pressure lowering effects [5,6,7], which shows that preserved eggs are beneficial for one’s health. After alkali curing, the proteins in the eggs degrade into small peptides and amino acids. This imparts a unique flavor and physiological benefits to the preserved eggs [5]. The unique gel network structure of the preserved eggs allows pepsin and pancreatic protease to bind more fully with it, speeding up the enzymatic digestion of the protease [8]. Therefore, preserved eggs have high digestive characteristics.

Digestion serves as a key step in the body’s metabolism and absorption of nutrients. Initially, researchers used a static digestion system with a simulated digestive solution and a thermostatic water bath to simulate in vivo digestive behavior [9]. This system was unable to realistically reproduce the human digestive environment, as well as the series of physical-mechanical movements and chemical reactions, limiting the in-depth study of digestive behavior and mechanisms [10]. Studies have been reported on dynamic digestive systems ranging from single static reactors to multi-chamber [11]. However, it is still a great challenge to reflect the digestive behavior in vivo more closely and realistically. An advanced ‘near-real’ dynamic human gastrointestinal-IV (DHGI-IV) system was developed by Chen et al. [12]. The DHGI-IV is designed in such a way that it simultaneously mimics the human digestive environment, the peristaltic junction of the stomach and intestines, and the fluid dynamic behavior in the digestive tract [13]. The DHGI-IV has been used in scientific studies for efficacy assessment of functional foods, drug delivery, infant formula optimization and the survival of intestinal probiotics [14,15,16,17]. In previous studies, the antioxidant, anti-inflammatory, anti-tumor and hypolipidemic effects of preserved eggs were investigated using an in vitro pepsin-pancreatic hydrolysis simulation model of gastrointestinal digestion [18,19,20]. However, the model failed to consider the influence of some factors such as the fold structure of the stomach and the continuous input of digestive juices. Therefore, it is more scientific and reasonable to use the DHGI-IV system to simulate food digestion.

It has been demonstrated that preserved eggs can promote Caco-2 apoptosis in tumor cells by up-regulating the expression of pro-apoptotic protein genes (caspase-3) [6]. Moreover, preserved egg white-derived peptides could up-regulate the expression of the anti-inflammatory gene interleukin-10 (IL-10) at a concentration of 0.25 mg/mL. IL-10 works as an antagonist of pro-inflammatory cytokine, and inhibits the development of tumorigenesis and leads to apoptosis [21]. Preserved eggs could also induce apoptosis in HT-29 and HepG2 cells by increasing the expression of the pro-apoptotic gene Bax and down-regulating the expression levels of the anti-apoptotic genes cyclooxygenase-2 (COX-2) and Bcl-2 [20]. A recent study showed that preserved eggs induced apoptosis in renal cancer cells by up-regulating the expression of the pro-apoptotic factors Bax and IL-10, and down-regulating the expression of the anti-apoptotic factors interleukin-4 (IL-4) and interleukin-6 (IL-6) [5]. Nevertheless, the anti-tumor mechanisms of the preserved eggs have not been fully elucidated and further studies are needed.

In this study, preserved eggs were placed in the DHGI-IV to simulate gastrointestinal digestion. Meanwhile, the pH, particle size, protein and fat digestibility of PED were measured, and their gastric-emptying properties and antioxidant activity were analyzed. Furthermore, we focused on the anti-tumor mechanism of PED in terms of cell apoptosis, migration, oxidative stress and angiogenesis using human hepatocellular carcinoma (HepG2). This has a positive effect on the developing resources for egg products, and also provides a theoretical basis for anti-tumor research on preserved eggs.

## 2. Materials and Methods

### 2.1. Materials and Reagents

The α-amylase from human saliva (Type IX-A, 1000–3000 units per mg protein) was obtained from Sigma-Aldrich, St. Louis, MO, USA. Pepsin from porcine gastric mucosa (≥3000 units per mg solid) and trypsin from porcine pancreas (≥250 units per mg solid) were purchased from Biosharp (Hefei, China). The Dulbecco’s modified eagle medium (DMEM) and fetal bovine serum (FBS) were purchased from HyClone (Cytiva, Emeryville, CA, USA). The 1% penicillin/streptomycin was obtained from Solarbio Science & Technology Co., Ltd. (Beijing, China). The CCK-8 kit and 5-Fluorouracil (5-FU) were provided by Beijing Zoman Biotechnology Co., Ltd. (Beijing, China). The 0.25% trypsin-EDTA, AO/EB and Hoechst 33342 stain were purchased Shanghai Yuanye Bio-Technology Co., Ltd. (Shanghai, China). PCR and cDNA reagents were obtained from ClonTech Takara^®^ Cellartis (Beijing, China). RNA extraction was performed using E.Z.N.A^®^ Total RNA kit 1 (Omega Bio Tek, Norcross, GA, USA). The Trizol reagent was purchased from Aidlab Biotechnologies Co., Ltd. (Beijing, China).

### 2.2. Gastrointestinal Digestive Properties of Preserved Eggs

#### 2.2.1. Preparation of Simulated Digestion Fluids

The simulated digestion fluids were prepared according to the experimental procedure reported by Minekus et al., with minor modifications made [22]. The compositions of simulated salivary fluid (SSF), simulated gastric fluid (SGF) and simulated intestinal fluid (SIF) stock solutions are shown in Appendix A. SSF consisted of SSF stock solution (diluted 1.25 times), 150 U/mL α-amylase [23] and pH adjusted to 7.0 ± 0.03 with 1 mol/L HCl. SGF which consisted of SGF stock solution (diluted 1.25 times), 4000 U/mL pepsin [24]. and pH was adjusted to 1.6 ± 0.03 using 6 mol/L HCl, which consisted of SIF stock solution (diluted 1.25 times), 200 U/mL trypsin, 20 mmol/L bovine bile salt [24], and pH adjusted to 7.0 ± 0.03 with 6.0 mol/L HCl.

#### 2.2.2. In Vitro Dynamic Digestion of Preserved Eggs

##### Oral Digestion

The preserved eggs were obtained from Hubei Shendan Health Food Co., Ltd. (Wuhan, China). They are generally produced by pickling fresh and intact duck eggs in a pickling solution containing 4.5% NaOH, 3.0% NaCl, 0.4% CuSO_4_ and 3.5% Chinese black tea at 25 °C for about 5 weeks [4]. The procedure for oral digestion of preserved eggs followed the experimental approach of Bellmann et al., with minor modifications made [25]. A total of 80 mL simulated salivary fluid (SSF) was placed in a 37 °C water bath for 5 min and then mixed with 200 g preserved eggs and 20 mL distilled water. The mixture was blended and beaten in a commercial electric meat mincer (HX-J3016, Foshan, China) for 1 min to simulate the mastication of preserved eggs.

##### The Dynamic Human Gastrointestinal-IV (DHGI-IV)

The DHGI-IV (Appendix A), the dynamic digestion model, was manufactured at Xiao Dong Pro-health Instrumentation Co. Ltd., Suzhou and was used in this study. All modules were housed in one aluminum cabinet. All operational parameters were pre-set through a touchscreen panel PLC control system. The DHGI-IV design concept is shown in Appendix A.

##### Gastrointestinal Digestion

To simulate a fasting state, 30 mL SGF was added to the stomach before the system was activated [26]. The simulation of oral digestion of preserved eggs sample was added to the bionic stomach of the DHGI-IV system through a funnel. The pneumatically-controlled driving DHGI-IV device was started immediately when the sample loading was completed, and relevant operating parameters are shown in Appendix A [17,23,27,28,29]. The PED were taken from the stomach and small intestine at 30 min intervals and their pH was immediately measured. The collected PED were also immediately heated at 100 °C for 10 min to inactivate enzymes. Afterwards, they were centrifuged at 4 °C, 5000 rpm for 10 min (Sigma3-30k, Sigma Aldrich, St. Louis, MO, USA) to obtain the supernatant for subsequent measurements.

#### 2.2.3. Determination of Protein and Fat Digestibility

##### Protein Digestibility

The protein contents of the PED (Section 2.2) were measured using the Pierce BCA Protein Assay Kit (Beyotime Biotechnology, Shanghai, China) in accordance with the instructions. The 50 μL PED were mixed with 4 mL of Coomasie Briliant Blue solution and were allowed to stand for 5 min. The absorbance value was measured at 595 nm with a UV spectrophotometer (Nanodrop2000/2000c; Thermo Scientific, Pittsburgh, PA, USA). Bovine serum protein was used as the standard, the protein digestibility was calculated according to the formula:(1)$$\mathrm{Protein}\;\mathrm{digestibility}\;\left(\%\right)=(1-[\left(A_{0}-A\right){/A}_{0}])\;\times \;100\;$$
where A_0_ and A (µg/mL) are the protein content of preserved eggs and PED, respectively.

##### Fat Digestibility

The crude fat contents of the PED (Section 2.2) were determined by the methods of the Association of Official Analytical Chemists (AOAC), with an automatic Soxhlet apparatus (SE-606, Jinan Alva Instruments Co. Jinan, China) following the manufacturer’s guidelines. The PED were lyophilized, packed in filter paper and then extracted with petroleum ether at 55 °C for 6 h. Upon completion of the fat extraction, the samples were dried at 60 °C for 12 h to remove residual water and petroleum ether. Sample fat digestibility was calculated as follow equation.
(2)$$\mathrm{Fat}\;\mathrm{digestibility}\;\left(\%\right)=(1-[\left(C_{0}\;-C\right){/C}_{0}])\;\times \;100$$
where C_0_ and C (g) are the fat contents of preserved eggs and PED, respectively.

#### 2.2.4. Determination Particle Size

The particle sizes of the PED (Section 2.2) were tested by MASTRERSIZER 2000 (Malvern Instruments Ltd., Worcestershire, UK). The refractive indices of PED were set to 1.333. The PED was poured into distilled water until the shading range (1–10%) was reached.

#### 2.2.5. In Vitro Gastric Emptying Assay of PED

During digestion, PED from the stomach were collected every 15 min, the volume was recorded, the volume was recorded, and the gastric emptying rate was derived from the liquid volume, solid volume and total solid-liquid volume. Furthermore, gastric retention rate data for PED were fitted by an optimized Elashoff’s power index model [30], as described in Appendix A.

#### 2.2.6. Determination of Antioxidant Activity

PED (Section 2.2) was lyophilized and prepared in distilled water to the appropriate concentration (62.5–2000 μg/mL). Subsequently, the scavenging activities of 2,2-Diphenyl-1-picrylhydrazyl (DPPH•), Hydroxy (•OH) and 2,2′-Azinobis-(3-ethylbenzthiazoline-6-sulfonate) radical (ABTS•) were evaluated [6,31,32]. Moreover, this paper also measured the efficacy of PED to reduce Fe^3+^ oxidation resistance (FRAP) [33]. Furthermore, the Trolox solutions (0–10 mmol/L) used as a standard solution and the results were expressed as Trolox equivalents (mmol/L). The test method is as described in Appendix A.

### 2.3. Effect of PED on HepG2 Cells

The PED (Section 2.2) was lyophilized and stored at 4 °C for anti-tumor activity studies.

#### 2.3.1. Cell Culture and Activity Assay

The HepG2 and human normal hepatocytes (HL-7702) cells were purchased from Wuhan Pro-cell (Wuhan, China) and were incubated in DMEM medium containing 10% FBS and 1% penicillin/streptomycin. After three passages, the cells were collected and inoculated onto 96-well plates. After overnight growth, fresh medium containing different concentrations of PED (62.5, 125, 250, 500, 1000, 2000, 4000 μg/mL) were added for incubation, with 5-FU (2000 μg/mL) as a positive control. After 24 h, the PED was incubated with CCK-8 reagent for 2 h and the absorbance was read at 450 nm by using an enzyme marker (iMark, Lenovo Biological Technology Co., Ltd., China). The survival rate was calculated according to the following formula:(3)Cell survival rate=[(As−Ab)/(Ac−Ab)
where As, Ab and Ac represent the absorbance of the sample, blank and control, respectively.

#### 2.3.2. Cell Morphological Observation

HepG2 cells were seeded in 6-well plates and were treated with different doses of PED. After 24 h, cells were stained with a dye mixture containing EB (100 μg/mL) and AO (100 μg/mL) at a 1:1 (*v/v*) for 5 min. Subsequently, the images of the cells were captured using a florescence microscope (OLYMPUS IX71, Tokyo, Japan). Furthermore, after treating the cells in the same way, Hoechst 33342 was added to stain the cells for 30 min and the nuclear changes were observed under a fluorescence microscope.

#### 2.3.3. Colony Formation Assay

The cells were inoculated onto 6-well plates at a density of 1 × 10^3^ cells/well and then incubated for 2 weeks to allow colony formation. After that, the cells were washed twice with PBS and fixed with 1% formaldehyde. After 30 min, staining was performed with crystal violet (0.1%, Sigma) containing 50% methanol and 10% glacial acetic acid. Images were observed by fluorescence microscopy and photographed for recording.

#### 2.3.4. Cellular Apoptosis Assay

The quantification of apoptosis was carried out using an Annexin V-FITC/PI kit (Biosharp, Hefei, China). Cell treatment was the same as in 2.3.2. PED treated cells were trypsinized and thereafter washed two times with PBS, and resuspended followed by the addition of binding buffer. Finally, the cells were incubated with 5 μL membrane linked protein V-FITC and 10 μL PI (20 μg/mL) stain in the dark for 30 min, according to the manufacturer’s instructions. Analysis was performed using a flow cytometer (BD FACSVerse™, Shanghai Weisu Biotech Co., Ltd. Shanghai, China).

#### 2.3.5. Cell Migration Examination

Logarithmic growth phase HepG2 cells were seeded in 6-well plates with 5 × 10^5^ cells. After 24 h, the medium was discarded and a straight line was scratched into each plate with the tip of a 200 μL gun. The cells were gently washed 3 times with PBS to remove cellular metabolites and detached cells, and then cells were treated with PED and continued incubation. The scratch was photographed at 0, 12 and 24 h by a microscope, and the relative speed of migration and closing areas were determined using Image J-Phase Wound Macro.

#### 2.3.6. Measurement of Intracellular ROS

Intracellular ROS production was measured with the ROS assay kit (Beyotime Biotechnology, Shanghai, China) with reference to the method of Marvibaigi et al. [34]. Briefly, the cells were seeded and incubated with different concentrations of PED for 24 h. The cells were then collected, washed twice with PBS and loaded with 1 mL 10 μM 2,7-dichlorodihydrofluorescein diacetate (H_2_DCFA-DA) for 30 min at 37 °C in the dark. Relative changes in intracellular ROS levels were monitored by flow cytometry.

#### 2.3.7. Real-Time PCR

First, HepG2 cells were treated with different concentrations of PED. After 24 h, total RNA was extracted from HepG2 cells using the chloroform-isopropanol method, in accordance with the Trizol kit instructions. Subsequently, RNA was quantified by Nano-Drop spectrophotometer, 1 μg RNA was taken and 20 μL reverse transcription system was used to synthesize cDNA, and then real-time quantitative PCR reactions were performed with SYBR Green Master, and each treatment was repeated three times. β-actin was used as an internal reference and the mRNA content was calculated by the 2^−ΔΔCt^ method. Reaction conditions: pre-denaturation at 95 °C for 10 min, denaturation at 95 °C for 15 s, annealing at 60 °C for 60 s, extension at 95 °C for 15 s, 40 cycles. The nucleotide sequences of the primers were designed by Wuhan Powerful Biotechnology Co., Ltd. (Appendix A).

#### 2.3.8. Western Blot Analysis

Cell treatment methods were the same as described above. After treatment, cells were collected, and total protein was extracted by adding RIPA lysate. The protein concentration was determined using a BCA kit. The samples were separated by SDS-PAGE (10%) gel and transferred to 0.45 μm PVDF membrane. Subsequently, a primary antibody (Bcl-2 (1:2000), Bak (1:1000), VEGF (1:1000), HIF-1α (1:1000) and β-actin (1:5000)) were added and incubated overnight at 4 °C, and then incubated with secondary antibodies for 2 h at room temperature. Blots were developed using the ECL detection system, and the relative density of immunoreactive bands were quantified by Image J software.

### 2.4. Statistics and Analysis

A one-way ANOVA test was used to compare the differences between means. Statistical analysis was performed by IBM SPSS software. Origin software was used to plot the graphs.

## 3. Results and Discussion

### 3.1. Changes in pH, Protein and Fat Digestibility of Preserved Eggs during Digestion

As shown in Figure 1A, the initial pH in the stomach was 1.6. After food intake, the magnitude of gastric pH depends on the secretion of gastric juice and the buffering effect of food [17]. During the first 30 min, the pH increased rapidly to 7.4. With the continuous secretion of SGF and the emptying of PED, the pH of the stomach continuously decreased. After 2 h of digestion, the pH in the stomach fluctuated around 2.0, basically reaching the fasting gastric juice state, and the trend was consistent with the results reported by in vivo experiments [17,35]. In addition, it was observed that the amount of residual food and digestive tract excretion of surimi in the gastric model was minimal (≤10% of the initial loading of the sample). The above results indicate that the samples were largely emptied after 2 h of digestion in the gastric model. The changes of pH during the intestinal digestion of PED are shown in Figure 1B. As the digestion reaction proceeded, the pH value of the PED showed a trend of increasing followed by decreasing. The pH increased in the early stages of digestion, as the sample was continuously digested by gastrointestinal peristalsis, extrusion, and the action of enzymes and bile. Subsequently, the pH decreased gradually as the samples were consumed. Overall, the pH range fluctuated between 7.01 and 8.39, which showed that the PED samples were still slightly alkaline after gastrointestinal digestion, in agreement with the findings of Mao et al. [6]. Studies have shown that alkaline foods (kelp, bitter melon and grapes) can effectively inhibit the growth of tumor cells [36,37,38].

As shown in Figure 1C,D, protein and fat digestibility increased as digestion progressed. After digestion was completed, the sample protein digestibility reached 90%. Furthermore, the fat digestibility increased rapidly in the first 1 h and levelled off as time increased, reaching 87% after 4 h. The optimal pH for the trypsin activity ranges between seven and nine [39]. The PED was slightly alkaline, and this led to an increase in trypsin activity and enhanced hydrolysis of proteins and fat. 

### 3.2. In Vitro Gastric Emptying Properties of PED

The process of emptying food from the stomach into the duodenum is called gastric emptying [40]. In the DHGI-IV, the gastric emptying curves for the PED mixture (PED, water and SGF), and solid (PED) and liquid fraction (water and gastric juice) are presented in Figure 2A. The parameters of *k*, *β*, *t*_1/2_ and *t_lag_*, obtained by fitting the in vitro gastric retention data with a modified Elashoff model, are shown in Appendix A. It could be seen that the liquid emptied in an approximately exponential manner throughout the 2 h of digestion and the solid emptied significantly slower than the liquid. The emptying pattern was similar to the reported gastric emptying curve in vivo [41]. During emptying, liquids and small particles flowed continuously from the stomach to the duodenum. In contrast, larger indigestible particles settled or accumulated in the stomach, and were later emptied continuously by the action of gastric acids and enzymes [40,42].

As shown in Figure 2A, after 45 min of digestion, there was almost no solid component in the collected digests. This indicates a delay in the evacuation of solid particles. The lag phase primarily reflects the time needed by the gastric antrum to reduce the ingested solid food into particles small enough to pass through the pylorus for emptying [43]. The lag phase can be adequately explained by Figure 2B. Anatomical features of the human stomach indicate that the stomach has a “J” shape. The particles with large mass fall at the bottom, as it is the lowest point of the stomach model, while small particles of material suspended in water fill the pyloric duct and empty more quickly [44].

### 3.3. Particle Size Distribution of PED

Figure 3 shows the particle size distribution of PED at different digestion times in the DHGI-IV. The mean median particle size (D_50_) indicates the pore size of the theoretical sieve through which 50% of the weight of the particles could pass [45]. The particle size of the samples decreased significantly as time progressed in the DHGI-IV. Specifically, D_50_ significantly decreased from an initial 447.7 μm to 133.9, 94.8, 53.3 and 42.3 μm after 1, 2, 3 and 4 h digestions, respectively (*p* < 0.05). Moreover, at the end of the digestion (t = 4 h), the particle size of the digestion products of the PED had a narrow and symmetrical dispersion relative to the percentages, with a more concentrated particle size distribution in the range of 15–100 µm. This indicated that the oils contained in the PED were well emulsified into smaller, more homogeneous droplets by the bile salts during gastrointestinal digestion [46]. These results indicated that the DHGI-IV showed good break down efficiency on the PED.

### 3.4. In Vitro Antioxidant Activity of PED

Preserved eggs contain a wider variety of protein and peptides that may have potentially strong biological functions [18]. Figure 4 shows the antioxidant activity of PED at different concentrations. As shown in the figure, the PED had a comparable outstanding capability to scavenge free radicals in a dose-dependent manner. PED significantly increased the free radical scavenging activity of ABTS, DPPH, FRAP and hydroxyl by about 15, 14, 10 and 8 times more than the control, respectively. Moreover, FRAP radical scavenging activity was the strongest (IC_50_ 3.51 mmol/L), followed by ABTS (IC_50_ 2.35 mmol/L) and hydroxyl (IC_50_ 1.13 mmol/L), but DPPH (IC_50_ 0.9 mmol/L) was the weakest within a certain concentration range of the PED. This suggested that the active groups were exposed after the digestion of the preserved eggs, and more active sites could act as hydrogen donors to inhibit oxidation reactions [6]. It has been shown that some active ingredients have the ability to significantly inhibit tumor cell proliferation while exerting antioxidant effects [47].

### 3.5. The Inhibition Effects of PED on Cell Growth

Persistent proliferation and multiplication are the basic characteristics of cancer cells [48]. As shown in Figure 5A, the PED significantly inhibited the proliferation of HepG2 cells in a dose-dependent manner with an IC_50_ value of 1555.78 μg/mL. Therefore, we selected the concentration ranges of PED from 250 to 1000 μg/mL for further experimental studies because of the relatively low toxicity of these doses. Meanwhile, we observed that PED had almost no effect on the viability of HL-7702 cells at the same concentration (Figure 5B). Moreover, the number of clones formed in HepG2 cells decreased significantly with increasing concentrations of PED treatment (Figure 5C,D). The rates of HepG2 clone formation in the low, medium and high dose groups were 56.14%, 35.61% and 23.64%, respectively, which were statistically significant compared to the control group (86.47%) (*p* < 0.05). The results showed that PED inhibited the proliferation of tumor cells HepG2, but had no effect on the growth of normal hepatocytes HL-7702.

### 3.6. The Induction of PED on Cell Apoptosis through the Mitochondrial Pathway

To observe the morphological changes of the HepG2 cells, we stained the cells with Hoechst 33342 and AO/EB. The characteristics of apoptosis include cell shrinkage, blebbing of the plasma membrane and chromatin condensation that are associated with DNA cleavage into ladders [49]. As demonstrated in Figure 6, the cells in the control group were green and showed normal morphology. After PED treatment, the early apoptotic cells showed a green solidified, or bead-like, shape due to the intact cell membrane and the solidified nucleus. However, the late apoptotic cells showed red solidified, or bead-like, shape due to the loss of cell membrane integrity and increased permeability. Furthermore, under the induction of PED, the nuclear morphology also changed. After the Hoechst 33342 staining, the nuclei of the control group were in the center of the cells, and the nuclei were uniform in size and light blue in color. The nuclei of the treated group showed lobed or bright blue fragmentation due to the concentration, and the nuclei became smaller and appeared condensed. With the increase in PED concentration, the change in cell morphology was more significant, and the cell density and number decreased more. 

In the early stage of apoptosis, the cell membrane is disrupted, leading to the transfer of phosphatidylserine (PS) outside the cell membrane and its selective binding to Annexin V. [50]. Next, the Annexin V/FITC assay was executed for apoptosis quantification. The effect of PED on apoptosis of HepG2 cells is shown in Figure 7, where Q1, Q2, Q3 and Q4 represent necrotic cells, early apoptotic cells, late apoptotic cells and living cells, respectively. As shown in Figure 7F, the apoptosis rate of control cells was 0.85%. Compared with the control, 24 h of cell exposure to PED caused a dose-dependent increase in total apoptotic cells, rising to 7.15%, 7.63% and 9.02% at 250, 500 and 1000 μg/mL concentrations, respectively. These data suggested that PED was able to induce apoptosis in HepG2 cells. The results suggested that preserved eggs could achieve anti-tumor effects by inducing apoptosis. 

Furthermore, the cell apoptotic was supported by PCR and western blotting assay. As shown in Figure 8A, the mRNA expression of the pro-apoptotic gene Bak was increased after PED exposure to HepG2 cells. The high-dose group was about 2.4 times higher than the control group, which was not significantly different from the 5-FU group. In contrast, the expression of the anti-apoptotic gene (Bcl-2) was down-regulated in a dose-dependent manner. In addition, the bands of Bcl-2 became lighter and the protein expression decreased after PED treatment, the corresponding activated Bak protein level increased and the bands became darker (Figure 8B,C).

The mitochondria apoptotic pathway is considered as one of the major intrinsic apoptotic pathways [51,52]. As an important regulator of apoptosis, the Bcl-2 protein family is mainly located in the mitochondria, endoplasmic reticulum and perinuclear membrane [53]. Moreover, the Bcl-2 protein family could be divided into two categories according to their structure and function: anti-apoptotic factors (Bcl-2, Bcl-xL, Bcl-W, Mcl-1, A1, Boo and Ced-9, etc.) and pro-apoptotic factors (Bax, Bak, Bim, Bad, Bid, Bik and Bok, etc.) [54,55]. Bak expression deregulation is associated with the development of many diseases and it is an important gene in the regulation of programmed cell death. It is generally accepted that a high expression of Bcl-2 inhibits apoptosis, whereas a high expression of Bak (Bax) promotes apoptosis [56]. The up-regulation and down-regulation of Bcl-2 and Bak expression suggest that the mechanism of PED-induced apoptosis in HepG2 cells is related to the mitochondrial apoptotic pathway. It was reported that preserved eggs could induce apoptosis in HT-29 and HepG2 cells by up-regulating the expression of Bax mRNA and down-regulating the expression of Bcl-2 mRNA [20].

### 3.7. The Induction of PED on Cell Apoptosis through Exacerbates Oxidative Stress

ROS are normal by-products of aerobic metabolism and they act as second messengers in various signal transduction pathways or in response to environmental stress. ROS generation is part of the mechanism by which most chemotherapeutic agents kill tumor cells [57]. It has been shown that cancer cell death is associated with high levels of ROS production, as high levels of ROS induced oxidative damage to biomolecules, including DNA, lipids and proteins [57,58]. Moreover, the role of high levels of ROS in inducing apoptosis in bladder cancer T24 cells was found in a study by Wang et al. [59]. To investigate the effect of PED on redox homeostasis in tumor cells, we examined intracellular ROS production. The results showed that the intracellular ROS content in HepG2 cells increased with the increase in PED treatment concentration. (Figure 9). The ROS levels in HepG2 cells were increased from 44.2% of untreated group to 48.2%, 98.6%, and 99.6% in the groups treated with 250, 500 and 1000 μg/mL of PED, respectively. It indicated that PED treatment could cause oxidative stress in HepG2 cells, which was contrary to the results in 3.4, and the exact reason for this needs to be further investigated. 

### 3.8. The Inhibition Effects of PED on Cell Migration and Angiogenesis-Related Gene Expression

The high metastatic and invasive capacity of tumors is the main characteristic of malignant tumors. More than 90% of deaths in people with cancer are due to the spread and metastasis of tumor cells, and it is one of the greatest challenges in cancer treatment [60]. Therefore, blocking the invasion and metastasis of tumor cells in all aspects is an effective means to inhibit tumor metastasis. As shown in Figure 10, the cells in the control group kept growing and multiplying and covered the scratches after 24 h. In contrast, cell migration was significantly inhibited after treatment with the PED. At different concentrations (250, 500 and 1000 µg/mL), cell migration was 78.67%, 55% and 8.33%, respectively, which were significantly lower than that of the control at 85.33% (*p* < 0.05). The study results showed that the PED could inhibit the migration of HepG2 cells.

Tumor growth depends on the formation of blood vessels in the tumor tissue, and after the tumor diameter exceeds 2–3 mm, it must rely on the tumor itself to form neovascularization to provide nutrients and oxygen [61]. Hepatocellular carcinoma is a vascular-rich tumor, and its occurrence and development cannot be separated from the formation of tumor neovascularization. Therefore, cutting off the feeding pathway for malignant tumor growth and blocking the bloodstream channels for tumor invasion and metastasis, which can effectively curb the malignant development of tumors [62]. HIF-1α is the only oxygen-regulated subunit that determines HIF-1α activity, and HIF-1α protein expression is increased in hypoxic environments. Moreover, it is involved in tumor angiogenesis by regulating the expression of multiple target genes and adapting to the hypoxic environment. VEGF is one of the target genes regulated by HIF-1α [63]. VEGF is a highly specific mitogen that directly stimulates neovascularization and is recognized as the most important and potent pro-angiogenic factor [64].

As shown in Figure 11, the relative mRNA and protein expressions of VEGF and HIF-1α were significantly reduced in a dose-dependent manner in cells from all PED concentration groups compared to the control group (*p* < 0.05). In addition, the results reported in Figure 11B,C showed that PED treatment resulted in a progressive decrease in the relative grey values of VEGF and HIF-1α and a significant concentration-dependent decrease in protein expression. The results showed that PED effectively inhibited tumor cell angiogenesis, which may be achieved by the reduction of intracellular HIF-1α expression, which mediates the reduction of VEGF expression. Numerous studies have confirmed the link between angiogenesis and hepatocellular carcinoma aggressiveness, and inhibition of angiogenesis significantly inhibited tumor growth and metastasis [65,66]. This also demonstrates that PED inhibits HepG2 cell migration.

## 4. Conclusions

The results of this study indicated that the preserved eggs were efficiently digested in the DHGI-IV system. After 4 h of digestion, protein and fat were significantly degraded, with 90% and 87% digestibility, respectively. The preserved eggs granularity was reduced by 9.6-fold. After 2 h of gastric digestion, preserved eggs were almost completely emptied with a lag time of 45 min. After digestion was completed, the pH of PED was 7.39. The study also revealed that the preserved eggs had significant antioxidant activity. PED effectively scavenged FRAP, ABTS, hydroxyl and DPPH radicals with IC_50_ of 3.51, 2.35, 1.13 and 0.9 mmol/L, respectively. At concentrations of 250–1000 μg/mL, PED significantly inhibited cellular activity, causing cellular crinkling and deformation and nucleus fragmentation. The ROS content of the high concentration group (1000 μg/mL) was twice as high as that of the control group. The high production of intracellular ROS led to the onset of apoptosis. Surprisingly, PED could effectively scavenge free radicals in vitro and had antioxidant effects. However, it induced a dramatic increase in intracellular ROS in tumor cells HepG2 and caused cellular oxidative stress. The reasons for this needs to be further studied. The down-regulation and up-regulation of apoptosis regulatory genes Bcl-2 and Bak expression activated the mitochondria-mediated apoptotic pathway. In addition, PED decreased the expression of angiogenic factors HIF-1α and VEGF, which inhibited cellular angiogenesis and migration. Further studies on preserved eggs should focus on the in vivo anti-cancer effects and the isolation of their anti-tumor active components.

## Figures and Tables

**Figure 1 foods-12-00800-f001:**
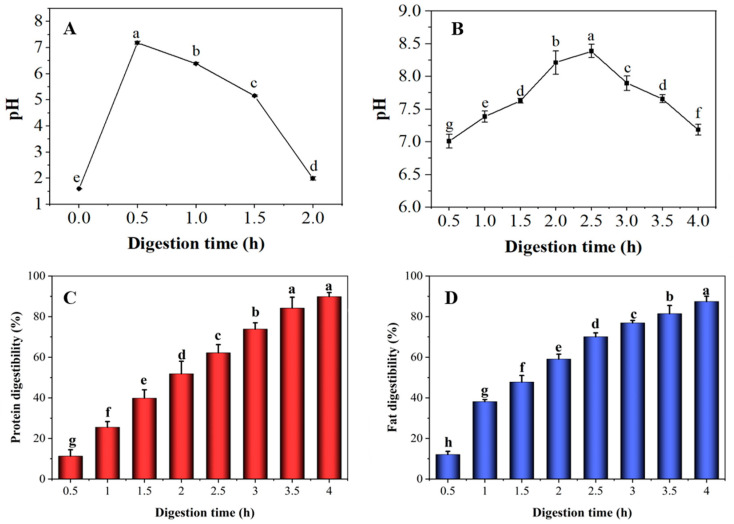
Changes in pH (**A**,**B**), protein (**C**) and fat (**D**) digestibility of preserved eggs during the digestion. **A**,**B** represent the pH of the stomach and intestine; **C**,**D** represent protein and fat digestibility during intestinal digestion. Different lowercase letters indicate significant differences (*p* < 0.05).

**Figure 2 foods-12-00800-f002:**
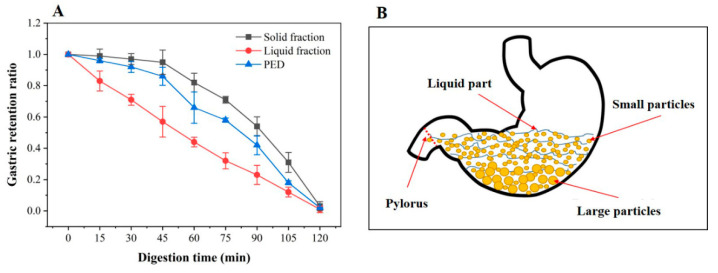
Gastric retention ratios (**A**) and the lagging phenomenon (**B**) of PED in the DHGI-IV model. PED: preserved egg digests; DHGI-IV: dynamic human gastrointestinal-IV.

**Figure 3 foods-12-00800-f003:**
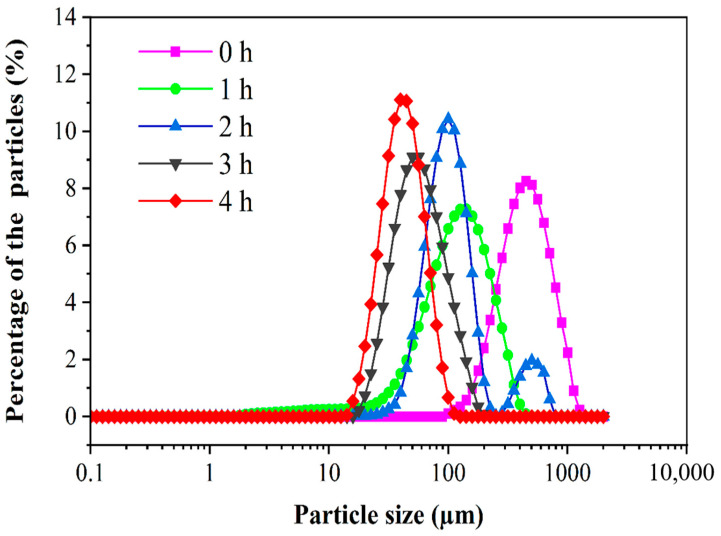
The size distributions of the PED at different digestion times in the DHGI-IV.

**Figure 4 foods-12-00800-f004:**
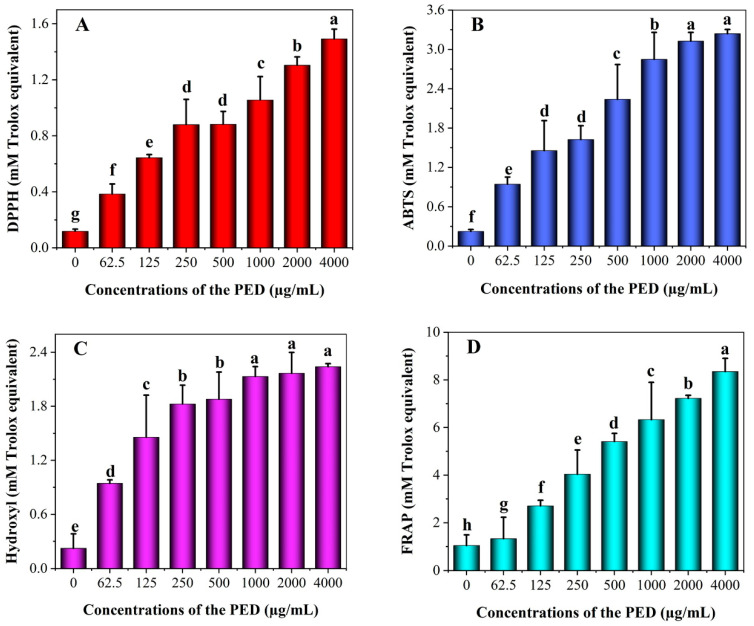
Antioxidant activity of PED in different concentrations (t = 4 h). (**A**): DPPH; (**B**): ABTS; (**C**): Hydroxyl; (**D**): FRAP. Bars indicate standard error (±SD). Different lowercase letters indicate significant differences (*p* < 0.05).

**Figure 5 foods-12-00800-f005:**
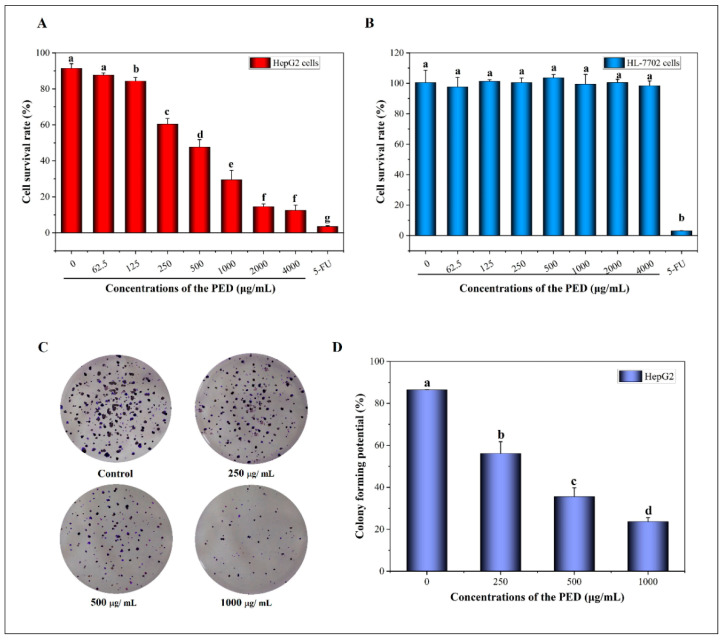
Effect of PED on the proliferation of HepG2 cells. (**A**,**B**): The HepG2 and HL-7702 cells survival rate; (**C**): Clone formation ability of HepG2 cells. (**D**): Clone formation rate the HepG2 cells. Different lowercase letters indicate significant differences (*p* < 0.05).

**Figure 6 foods-12-00800-f006:**
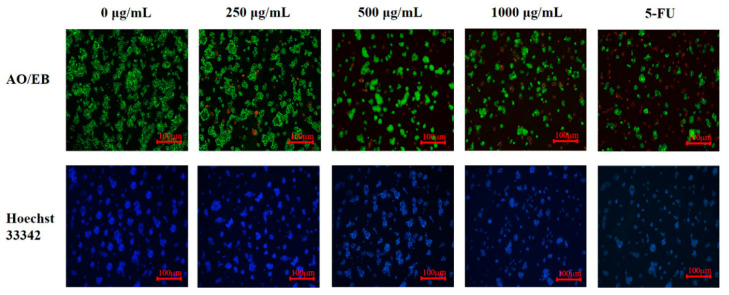
Effect of PED on the cell morphology of HepG2 cells.

**Figure 7 foods-12-00800-f007:**
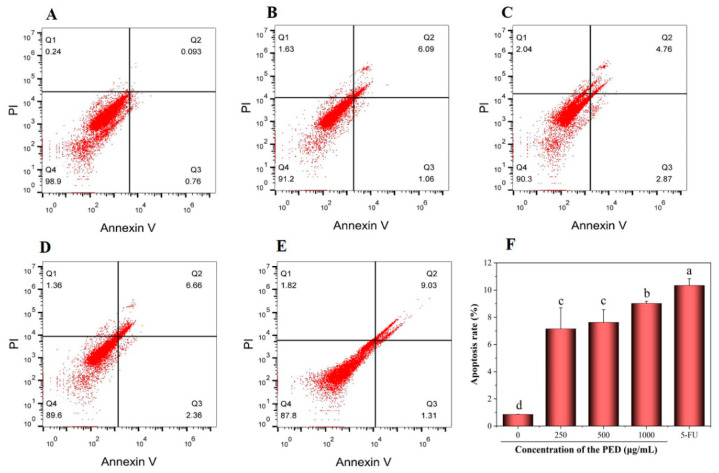
Effect of different doses of PED on apoptosis of HepG2 cells. (**A**): untreated cells; (**B**–**D**): 250, 500, and 1000 μg/mL; (**E**): 5-FU; (**F**): Analysis of apoptosis rate of HepG2 cells. 5-FU: 5-Fluorouracil. Different lowercase letters indicate significant differences (*p* < 0.05).

**Figure 8 foods-12-00800-f008:**
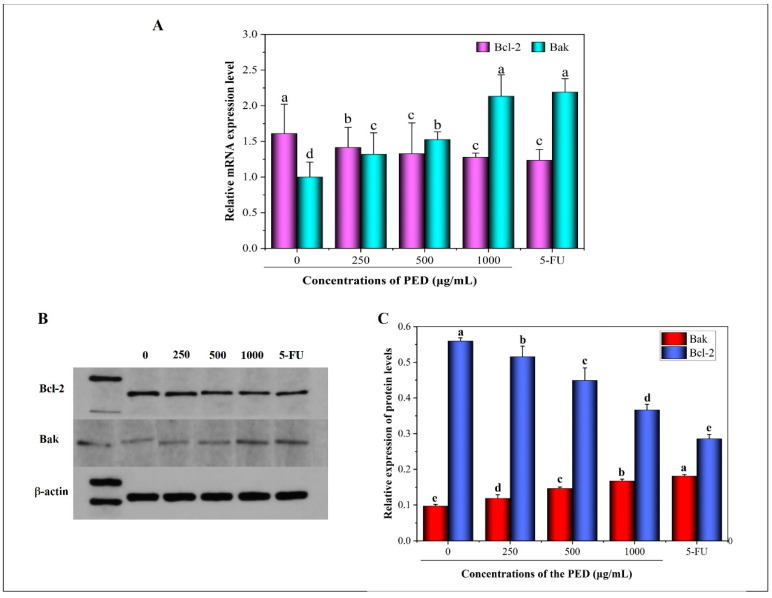
Effect of PED on the expression levels of Bcl-2 and Bak. (**A**): the mRNA expression levels; (**B**,**C**): the protein expression levels. The β-actin was indicated the internal control. Different lowercase letters indicate significant differences (*p* < 0.05).

**Figure 9 foods-12-00800-f009:**
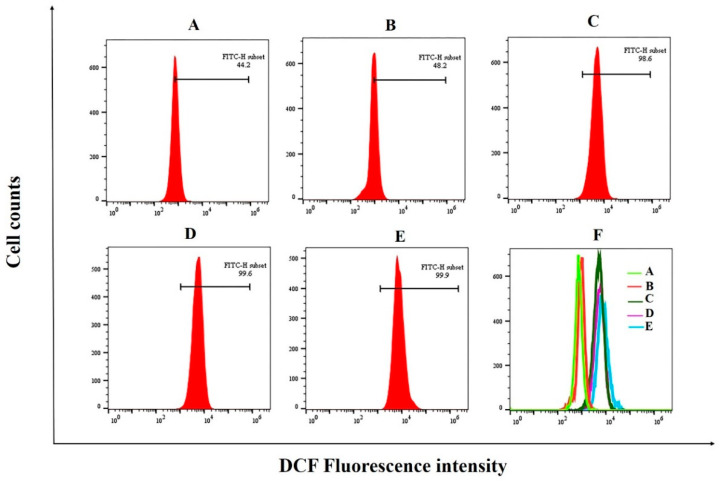
Effect of PED treatment on ROS generation in HepG2 cells. (**A**): untreated cells; (**B**–**D**): 250, 500, and 1000 μg/mL; (**E**): 5-FU; (**F**): ROS generation profiles under treatment with different concentrations of PED.

**Figure 10 foods-12-00800-f010:**
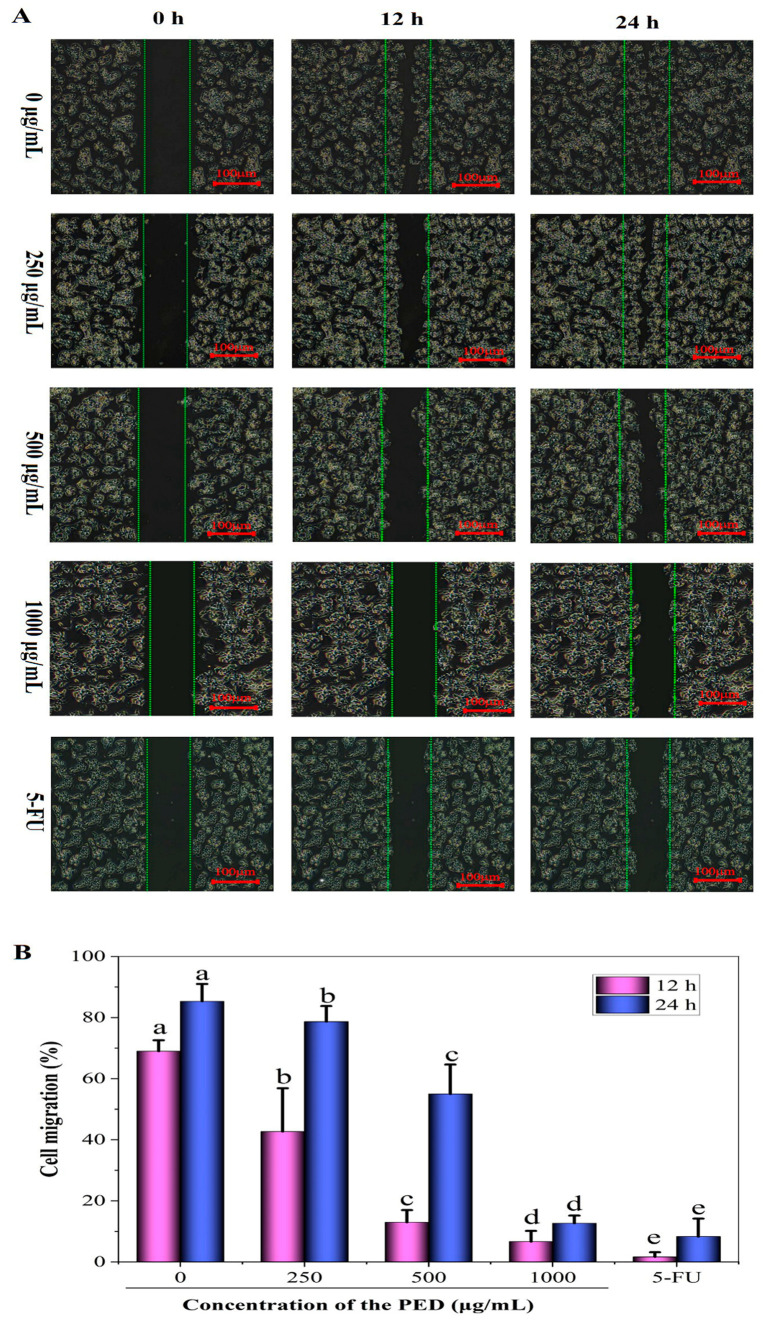
Effect of PED on the cell migration of HepG2 cells. (**A**) Images of PED-treated HepG2 cells monolayers at different time intervals (0, 12 and 24 h). (**B**) Quantitative measurement of HepG2 cells migration after treatment with PED. Different lowercase letters indicate significant differences (*p* < 0.05).

**Figure 11 foods-12-00800-f011:**
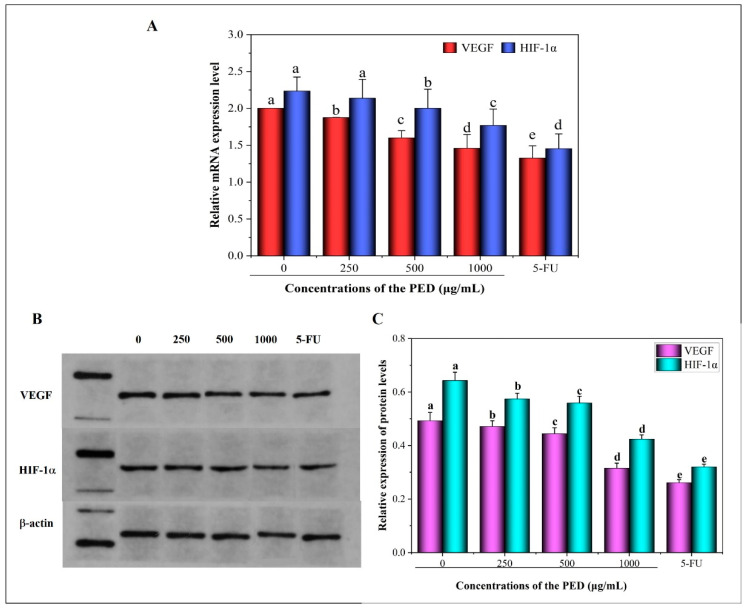
Effect of PED on the expression levels of HIF-1α and VEGF. (**A**): the mRNA expression levels; (**B**,**C**): the protein expression levels. The β-actin was indicated the internal control. Different lowercase letters indicate significant differences (*p* < 0.05).

## Data Availability

Data are available on request from the authors.

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
