# Peer review of "Characterization of the Dynamic Gastrointestinal Digests of the Preserved Eggs and Their Effect and Mechanism on HepG2 Cells"

_foods, 2023, doi:10.3390/foods12040800_

Round 1

Reviewer 1 Report

The article is well written and organized, the only comment or suggestion is: since there are some articles published on the effect of preserved eggs on HepG2 cells, the authors should be more specific about the results found that have not been reported. published and, that justify the publication of the article.

Some articles to considere are:

PAN J, LIANG Y, ZHANG M, LIU X, ZHAO S, HUANG S, JIN G. 2022. Effect of Preserved Eggs Simulated Gastrointestinal Digests on HepG2 Cells[J]. Journal of Food Science and Technology,2022,40(2):82-90.

Yanhui L, Lichao H, Min Z, Xiaojie L, Guofeng J, Yongguo J, Meihu M.2020

Preserved egg digests promote the apoptosis of HT29 and HepG2 cells,

Food Bioscience. 36,100661. https://doi.org/10.1016/j.fbio.2020.100661.

Author Response

Responses to the reviewer' comments/suggestions and the changes made in the revised manuscript

Dear Reviewer:

Thank you very much for your evaluation and comments on our paper “Characterization of the dynamic gastrointestinal digests of the preserved eggs and their effect and mechanism on HepG2 cells”. (Foods - 2092320). The manuscript has been revised according to the your constructive and valuable comments. We have carefully considered all your suggestions and answered the questions accordingly. The main corrections in the paper and the responds to your comments are as flowing:

Reviewer' comments:

The article is well written and organized, the only comment or suggestion is: since there are some articles published on the effect of preserved eggs on HepG2 cells, the authors should be more specific about the results found that have not been reported. published and that justify the publication of the article.

Some articles to consider are:

PAN J, LIANG Y, ZHANG M, LIU X, ZHAO S, HUANG S, JIN G. 2022. Effect of Preserved Eggs Simulated Gastrointestinal Digests on HepG2 Cells[J]. Journal of Food Science and Technology,2022,40(2):82-90.

Yanhui L, Lichao H, Min Z, Xiaojie L, Guofeng J, Yongguo J, Meihu M.2020 Preserved egg digests promote the apoptosis of HT29 and HepG2 cells, Food Bioscience. 36, 100661. https://doi.org/10.1016/j.fbio.2020.100661.

Response: Thank you for your guidance. In our study, we used a dynamic gastrointestinal simulation digestion system (dynamic human gastrointestinal-IV (DHGI-IV)) for the digestion of preserved eggs. The pH, particle size, protein/fat digestibility, gastric emptying characteristics and antioxidant activity of the digested samples were also analyzed, with a focus on their anti-tumor function. Unlike past publications, we investigated the effects of preserved eggs digestion samples on HepG2 cells in terms of cell migration, oxidative stress, expression of apoptosis-related proteins and angiogenic proteins. The results showed that the preserved egg digestion samples inhibited cell migration and the expression of angiogenic proteins HIF-1a and VEGF. It also induced cellular oxidative stress, regulated the expression of apoptosis proteins Bcl-2 and Bak, and induced apoptosis.

Thank you and best regards.

Yours sincerely,

Meihu Ma

Reviewer 2 Report

To author,

In my opinion, the subject is very interesting and important. The study is a novel source of bioactive peptide from egg products, and the research is within the scope of the journal. However, the manuscript needs some minor modifications and clarifications, which need to be taken care of, and the paper needs to be modified. Following are my specific questions/and general comments;

1)     Line 102: The process of producing preserved eggs should be explained more thoroughly.

2)     Line 135: The formula might be "Protein digestibility (%) = ( 1 – [(A0-A)/A0]) ×100"

3)     Line 135: The formular might be "Fat digestibility (%) = ( 1 – [(C0-C)/C0] )×100"

4)     Line 159: "concentration (62.5-2000 μg/mL)", this concentration should be indicated; is it μg/L of protein or peptide?

5)     Line 267: It’s not clear in Figure 1. "Changes in pH (A-B)", what is the difference between these two pictures?

6)     Lines 319–321: The reference to Tan et al. did not relate to the sentence, "The results of Tan et al. showed that PED could effectively scavenge ABTS and FRAP free radicals and could be used as a source of antioxidants in the food industry."

7)     Conclusions: It seems to me that the conclusions are consistent with the results of the investigation. It is better to add more discussion about the appropriate concentration for application.

Author Response

Responses to the reviewer' comments/suggestions and the changes made in the revised manuscript

Dear Reviewer:

Thank you very much for your evaluation and comments on our paper “Characterization of the dynamic gastrointestinal digests of the preserved eggs and their effect and mechanism on HepG2 cells”. (Foods - 2092320). The manuscript has been revised according to the your constructive and valuable comments. We have carefully considered all your suggestions and answered the questions accordingly. The main corrections in the paper and the responds to your comments are as flowing:

Reviewer’s comments:

In my opinion, the subject is very interesting and important. The study is a novel source of bioactive peptide from egg products, and the research is within the scope of the journal. However, the manuscript needs some minor modifications and clarifications, which need to be taken care of, and the paper needs to be modified. Following are my specific questions/and general comments;

  • Line 102: The process of producing preserved eggs should be explained more thoroughly.

Response: Thanks for your valuable advice. Lines 115-117, the production process of preserved eggs has been added.

  • Line 135: The formula might be "Protein digestibility (%) = ( 1 – [(A0-A)/A0]) ×100"

Response: Thank you for your guidance. Line 149, this formula has been modified to "Protein digestibility (%) = (1 - [(A0-A)/A0]) × 100".

  • Line 135: The formula might be "Fat digestibility (%) = ( 1 – [(C0-C)/C0] )×100".

Response: Thank you for your guidance. Line 160, this formula has been modified to "Fat digestibility (%) = (1 – [(C0-C)/C0]) × 100".

  • Line 159: "concentration (62.5-2000 μg/mL)", this concentration should be indicated; is it μg/L of protein or peptide?

Response: Thank you for your guidance. Line 174, "Concentration (62.5-2000 μg/mL)", is not a protein or peptide, but preserved egg digests (PED).

  • Line 267: It’s not clear in Figure 1. "Changes in pH (A-B)", what is the difference between these two pictures?

Response: Thank you for your guidance. In Figure 1 on line 285, A and B indicate the pH changes of the preserved eggs for digestion in the stomach and intestine, respectively. It has been marked for clarification on line 286.

  • Lines 319–321: The reference to Tan et al. did not relate to the sentence, "The results of Tan et al. showed that PED could effectively scavenge ABTS and FRAP free radicals and could be used as a source of antioxidants in the food industry."

Response: Thank you for your guidance. Lines 319-321: the reference to Tan et al. has been removed.

  • Conclusions: It seems to me that the conclusions are consistent with the results of the investigation. It is better to add more discussion about the appropriate concentration for application.

Response: Thank you for your guidance. More discussion on the appropriate concentration to apply has been added to the conclusions.

Thank you and best regards.

Yours sincerely,

Meihu Ma

Reviewer 3 Report

Reviewer suggestion:

The manuscript explains the Characterization of the dynamic gastrointestinal digests of the preserved eggs and their effect and mechanism on HepG2 cells for the development of functional food.  But we hope that the quality of the manuscript could be enhanced based on these several suggestions :

1.      Abstract

Please describe more about the quantitative data and also the main finding of the research

At abstract should mention more not just the introduction about the cultivar effect but also the introduction about the shading effect

2.     Introduction

Please kindly added the novelty of the research in the introduction

Please mention the pros and cons using simulative digestion rather than in vivo assay

It seems that the aims of the research is not align, on the introduction the sate of the art of aiming anticancer activity is not mentioned

3.     Material and Methods

1.     Kindly describe the sample, how to make the preserved eggs?

2.     Komas Briliant Blue ?  Is it Coomasie Briliant Blue?

3.     How to determine anticancer activity ?

4.     Results and Discussion

1.     Please add more discussion on each parameter.

5.      Conclusions

Please add more comprehensive conclusions for the main findings of the research

Author Response

Responses to the reviewer' comments/suggestions and the changes made in the revised manuscript

Dear Reviewer:

Thank you very much for your evaluation and comments on our paper “Characterization of the dynamic gastrointestinal digests of the preserved eggs and their effect and mechanism on HepG2 cells”. (Foods - 2092320). The manuscript has been revised according to the your constructive and valuable comments. We have carefully considered all your suggestions and answered the questions accordingly. The main corrections in the paper and the responds to your comments are as flowing:

Reviewer' comments:

The manuscript explains the Characterization of the dynamic gastrointestinal digests of the preserved eggs and their effect and mechanism on HepG2 cells for the development of functional food. But we hope that the quality of the manuscript could be enhanced based on these several suggestions:

  1. Abstract
    • Please describe more about the quantitative data and also the main finding of the research.

Response: Thanks for your valuable advice. The quantitative data as well as the main findings of the study have been described in the abstract.

  • At abstract should mention more not just the introduction about the cultivar effect but also the introduction about the shading effect.

Response: Thank you for your guidance. All the effects, including cultivar effects and shading effects, have been described in the abstract.

  1. Introduction
    • Please kindly added the novelty of the research in the introduction

Response: Thank you for your guidance. The novelty of the study has been added in the introduction.

  • Please mention the pros and cons using simulative digestion rather than in vivo assay

Response: Thank you for your guidance. The pros and cons using simulated digestion have been described.

  • It seems that the aims of the research is not align, on the introduction the sate of the art of aiming anticancer activity is not mentioned

Response: Thank you for your guidance. In the introduction, the current state of the art in anti-hepatocellular carcinoma is described.

  1. Material and Methods
    • Kindly describe the sample, how to make the preserved eggs?

Response: Thanks for your valuable advice. Lines 115-117, the production process of preserved eggs has been added.

  • Komas Briliant Blue? Is it Coomasie Briliant Blue?

Response: Thank you for your guidance. Line 144 "Komas Briliant Blue" has been changed to "Coomasie Briliant Blue".

  • How to determine anticancer activity ?

Response: Thank you for your guidance. The survival rate of HepG2 cells was used to indicate the anti-tumor activity of the samples. The lower the cell viability, the stronger the antitumor activity. The cell viability formula is shown in line 194.

  1. Results and Discussion
    • Please add more discussion on each parameter.

Response: Thanks for your valuable advice. The results of the data have been discussed more in the Results and Discussion section.

  1. Conclusions
    • Please add more comprehensive conclusions for the main findings of the research.

Response: Thank you for your guidance. The main findings of the study have been further summarized.

Thank you and best regards.

Yours sincerely,

Meihu Ma

Round 2

Reviewer 3 Report

The revised version already answered our questions and also have a lot of improvement